# β3 Receptor Signaling in Pregnant Human Myometrium Suggests a Role for β3 Agonists as Tocolytics

**DOI:** 10.3390/biom13061005

**Published:** 2023-06-17

**Authors:** Iain L. O. Buxton, Hazik Asif, Scott D. Barnett

**Affiliations:** Myometrial Function Group, University of Nevada, Reno School of Medicine, Reno, NV 89557, USA

**Keywords:** pregnancy, myometrium, preterm labor, β3 adrenergic receptor, smooth muscle, tocolysis

## Abstract

Preterm labor leading to preterm birth is the leading cause of infant morbidity and mortality. At the present time, nothing can reliably halt labor once it begins. The knowledge that agonists of the β2 adrenergic receptor relax airway smooth muscle and are effective in the treatment of asthma led to the notion that β2 mimetics would prevent preterm birth by relaxing uterine smooth muscle. The activation of cAMP-dependent protein kinase by β2 receptors is unable to provide meaningful tocolysis. The failure of β2 agonists such as ritodrine and terbutaline to prevent preterm birth suggests that the regulation of uterine smooth muscle is disparate from that of airway. Other smooth muscle quiescent-mediating molecules, such as nitric oxide, relax vascular smooth muscle in a cGMP-protein kinase G-dependent manner; however, nitric oxide activation of protein kinase G fails to explain the relaxation of the myometrium to nitric oxide. Moreover, nitric oxide-mediated relaxation is blunted in preterm labor, and thus, for this reason and because of the fall in maternal blood pressure, nitric oxide cannot be employed as a tocolytic. The β3 adrenergic receptor-mediated relaxation of the human myometrium is claimed to be cAMP-dependent protein kinase-dependent. This is scientifically displeasing given the failure of β2 agonists as tocolytics and suggests a non-canonical signaling role for β3AR in myometrium. The addition of the β3 agonist mirabegron to pregnant human myometrial strips in the tissue bath relaxes oxytocin-induced contractions. Mirabegron stimulates nitric oxide production in myometrial microvascular endothelial cells, and the relaxation of uterine tissue in vitro is partially blocked by the addition of the endothelial nitric oxide synthase blocker Nω-Nitro-L-arginine. Recent data suggest that both endothelial and smooth muscle cells respond to β3 stimulation and contribute to relaxation through disparate signaling pathways. The repurposing of approved medications such as mirabegron (Mybetriq™) tested in human myometrium as uterine tocolytics can advance the prevention of preterm birth.

## 1. Introduction

Preterm birth (PTB), defined as delivery of a fetus prior to 37 weeks of completed gestation, is a global problem, and premature babies are at risk of major disabilities [1]. Preterm labor (PTL) is the proximate cause of PTB [2]. The incidence of PTB is troubling with the overall incidence at about one in ten pregnancies [3]. African American women have about a 40% increased chance to deliver preterm [4], a recognized health disparity [5]. While there are many variables associated with PTL incidence, about half of all PTL cases are spontaneous (idiopathic) in nature [2]. Our lack of a complete understanding of the events that lead to normal-term labor have shrouded spontaneous prematurity in mystery. In the United States, obstetricians have few tools available to manage a woman who enters labor too soon. No medication enjoys an FDA-approved indication to prevent labor. Drugs employed as tocolytics represent *borrowed pharmacology*, that is, medications such as the β2 adrenergic agonists terbutaline/salbutamol, or the calcium channel blocker nifedipine often employed to delay labor, are administered based on their actions in airway and vascular smooth muscle, respectively, and are not targeted based on their efficacy to prevent uterine contractions during labor. 

Tocolytics currently in use have unwanted effects on the mother and fetus that limit dosing. This leaves open the question as to the possible efficacy of drugs, such as nifedipine, were higher-dose administration possible and suggests that combination tocolysis targeting more than one pathway might provide effective tocolysis. Efforts in the 1980s to develop effective tocolytics targeted to act in the myometrium resulted in the development of atosiban (Tractocile™), an antagonist of the oxytocin receptor [6]. Despite the logic behind its development, atosiban has not proven effective in preventing PTB [7] and is not approved for use as a tocolytic in the United States. Even the use of 17-hydroxyprogesterone caproate (Makena™), thought to be beneficial as a tocolytic in patients with a short cervix, has been withdrawn for lack of efficacy [8].

### Current Tocolytic Management

When a patient presents in labor preterm before 31–32 weeks of completed gestation, treatment with a non-steroidal antiinflammatory, indomethacin or ketorolac, may be initiated orally for 48 h [9]. Oral nifedipine may also be tried for 48 h and is sometimes combined with magnesium sulfate, thought to provide neuroprotection [10]. When employed, the β2 adrenergic agonist terbutaline is sometimes viewed as a triage medication to identify responders. Despite the varied experience of obstetricians and some claims of short-term benefit, tocolytic therapy is unable to reliably prevent labor once it begins [11].

Our laboratory is focused on understanding the regulation of uterine quiescence. In order to accomplish this goal, it is necessary to understand the unique pharmacology of the pregnant myometrium. Rather than attempting to block signaling pathways classically associated with smooth muscle contraction, we suggest that describing targets that regulate quiescence in the pregnant uterus will serve the development of novel tocolytics. To this end, we are pursuing the repurposing of existing medications that act in the myometrium to affect pathways known to be dysfunctional in preterm myometrium. Examining the efficacy of drugs that relax the pregnant human uterine smooth muscle can provide evidence of what agents in combination might overcome early labor. Here, we describe a role for β3 adrenergic receptor agonists in tocolytic drug development.

## 2. Dysregulated Pathways in Myometrium

### 2.1. Calcium

Much of what we think we know about uterine smooth muscle cell signaling regulation is based on vascular or gastrointestinal smooth muscle. While this is obviously important, we emphasize that myometrium is a unique smooth muscle with disparate regulation of pathways leading to quiescence. To govern Ca^2+^-dependent functions in reaction to disparate stimuli, uterine smooth muscle employs plasmalemmal and sarcoplasmic reticulum (SR) Ca^2+^ channels to produce oscillating Ca^2+^ signals, which can differ in their spatial and temporal distribution [12,13]. These signals include global changes in [Ca^2+^]i as well as highly localized Ca^2+^ release events (e.g., sparklets and sparks) [13]. Calcium can enter the cell from the extracellular space or be released from the intracellular Ca^2+^ SR store, or both. Extracellular Ca^2+^ influx is principally mediated by voltage-dependent L-type Ca^2+^ channels (LTCCs) that are targeted by nifedipine. Because of their high single-channel conductance, LTCCs have the largest influence on global [Ca^2+^]i that determines the contractile state of the uterine muscle and ultimately parturition [14]. Nonetheless, neither nifedipine, which blocks these channels, nor MgSO_4_, which reduces Ca^2+^ availability in the cell, provide adequate tocolysis [15,16,17] at safe doses. 

Moreover, the blockade of Ca^2+^ entry alone as a measure of tocolysis does not fit with our understanding of the compartmentation of calcium signaling [18]. For example, activation of the recently discovered inwardly rectifying Ca^2+^-channel Piezo1 in myometrium does not result in myometrial contraction as might be expected of Ca^2+^ entry. Indeed, activation of Piezo1 by the selective agonist yoda1 relaxes the pregnant myometrium [19]. Calcium also regulates the outward movement of potassium for uterine muscle thought to mediate relaxation. The calcium-activated K^+^ channel (BK_Ca_), the most abundant K^+^ channel in myometrium [20], is composed of homo-tetramers of α subunits, each comprising seven conserved transmembrane domains, an extracellular N-terminus, and a large C-terminal domain that regulates K+ conductance and Ca^2+^ sensing. The α subunit forms the channel pore and is associated with a β1 subunit expressed in human myometrium [21]. Lorca et al. [22] laid out the argument for the importance of BK_Ca_ as a key regulator of membrane potential and uterine quiescence based on abundance [20], activation by Ca^2+^ influx [23], inhibition that depolarizes and increases myometrial contractility, and BK_Ca_ channel opening that relaxes myometrium [24]. Both BK_Ca_ channel activators, such as the benzimidazolone NS1619 [25], and inhibitors such as paxilline are available [26]. 

### 2.2. NO Signaling Exception

During gestation, the myometrium must remain relaxed to accommodate the growing fetus. While many studies have focused on the regulation of contraction per se, recent efforts are focused on the disparate nature of myometrial quiescence. These efforts have uncovered a unique aspect of myometrial quiescence regulation: its dysfunctional response to nitric oxide (NO)-mediated relaxation in human spontaneous PTL tissues correlated with the overexpression of S-nitrosoglutathione reductase (GSNOR) that metabolizes NO [27]. We have proposed that differences in posttranslational protein modification by S-nitrosation underlies the dysfunctional PTL response to NO in preterm tissues [27]. 

The distinct physiology of myometrium is evident from our discovery that NO does not relax the myometrium in a cyclic guanosine monophosphate (cGMP)-dependent manner. Canonically, NO activates guanylyl cyclase, and cGMP is generated and activates its cognate kinase, protein kinase G (PKG). PKG then phosphorylates proteins to regulate their function [28]. In smooth muscle, the NO-cGMP-PKG pathway is expected to lead to the phosphorylation of myosin light-chain phosphatase to dephosphorylate myosin and reduce cross-bridge cycling leading to relaxation. Unlike vascular muscle, however, term myometrium relaxes to NO despite a blockade of cGMP accumulation [29] or the over-accumulation of cGMP from non-receptor-mediated activation of guanylyl cyclase by cinaciguat (BAY58-2667) [30]. While PKG is activated in these experiments and proteins are phosphorylated, relaxation does not result [28]. These data clearly established a dissociation of cGMP accumulation and myometrial relaxation that is a unique exception to Nobel dogma. NO-mediated relaxation of pregnant human myometrium (EC_50_, 1 μM), together with its cGMP independence [31], established that another mechanism mediates relaxation to NO [32,33]. While unexpected, these findings challenged the dogma and emphasized the unique nature of human myometrium. 

### 2.3. S-Nitrosation

S-nitrosation describes a thiol (e.g., cysteine) converted to a nitrosothiol (S-NO) by a one-electron oxidation from the NO radical [34]. While many descriptions of the addition of a NO group to a protein cysteine employ the term nitrosylation, this term is best used to distinguish nitrosation of a metal-centered protein such as guanylyl cyclase [35]. We use the term nitrosation to describe the addition of a nitro group to contraction/relaxation-associated proteins in uterine muscle. Protein S-nitrosations are not enzymatically mediated, resulting in a labile posttranslational modification that is moderated by the stoichiometric availability of NO and protein conformations that present with cysteines. 

The absence of nitrergic nerves in the pregnant human myometrium raises the question as to whether NO acts endogenously to regulate uterine contraction. The notion that the source of NO is the smooth muscle of the uterus itself does not hold for human uterine muscle cells [36], although sources of NO that can be available to regulate the uterine smooth muscle are several. NO may be carried as S-nitrosoglutathione [37] (GSNO) from microvascular endothelium [34], syncytiotrophoblasts [38], macrophages [39], and placenta [40,41] to the myometrium. Myometrial proteins are known to be differentially S-nitrosated in term vs. preterm myometrium in a pregnancy-state-specific manner [42]. NO-mediated relaxation of term myometrium likely results from S-nitrosation of proteins involved in contraction–relaxation signaling because these proteins are differentially S-nitrosated in PTL. This underscores the point that the nitrosation pathway is regulated and not obligatory. Examination of the S-nitrosation pathway in a targeted fashion may represent the intersection of numerous potential initiators of PTL and represents an important area of study. 

### 2.4. Nitric Oxide Synthase

In the myometrium, endothelial NO synthase (eNOS) is expressed in microvascular endothelial cells and not in uterine smooth muscle cells. Careful studies of our own some years ago, and more recently by Bartlett et al., revealed that no form of NOS is expressed in the uterine muscle cell [36]. Norman et al. suggested that the inducible form of NOS, iNOS, was present in pregnancy samples, but no cellular breakdown was made [43], and while several studies have reported iNOS and even neuronal NOS in myometrium from animal models [44], this is not the case for human uterine smooth muscle. Thus, the observation of β3 adrenergic receptor-mediated NO availability can be seen as a paracrine-related event from a non-myocyte cell type. In the human uterus, this cell type is the microvascular endothelial cell within the muscle [45]. 

The activity of eNOS is affected by its phosphorylation [46], acetylation [47], and S-nitrosation [48]. Under basal conditions, eNOS is anchored to the sarcolemmal and transverse-tubule plasma membranes by acylation and is maintained in an inactive state through its interaction with the scaffolding domain of caveolin [49]. The targeting of eNOS to the plasma membrane results in constitutive S-nitrosation of eNOS, and following β3 adrenergic receptor agonist stimulation, eNOS is de-anchored from the membrane, and an increase in the level of intracellular Ca^2+^ leads to the disruption of the caveolin–eNOS inhibitory interaction followed by the recruitment of Src kinase [50]. Src activates eNOS by phosphorylation of tyrosine-83 (Y83). The activated enzyme then translocates to the cytosolic fraction.

Despite the dogma surrounding the actions of NO [51], in the case of myometrium, it is intellectually displeasing to propose that NO itself traverses multiple membrane barriers to S-nitrosate specific proteins. The S-nitrosation of contraction-associated proteins seen in term versus preterm myometrium is different [42]. We suggest that NO is carried as, or immediately buffered as, S-nitrosoglutathione (GSNO) across membranes to enter the myocyte. NO from the endothelium (and putatively syncytiotrophoblasts [38], macrophages [39], and placenta [40,41]) acts on the myometrium where proteins are differentially S-nitrosated in term versus preterm myometrium in a pregnancy-state-specific manner [42]. Excessive inflammation (oxidative stress), often cited as a contributor to PTL [39], decreases S-nitrosations [52], suggesting that inflammation associated with PTL may act through disruption of the S-nitrosation pathway.

Our laboratory was the first to directly demonstrate the failure of NO to relax PTL myometrium no matter the concentration employed, an observation that was validated by our discovery of the overexpression of GSNOR, the predominate metabolizer of NO in the preterm myometrium [27]. Thus, while NO donors cannot be useful for preventing PTL, the differences in the effect of NO on the uterine smooth muscle from term versus preterm patients suggests that there are distinct differences in the S-nitrosation pathway in early myometrium in certain patients, and this difference may reveal new targets for tocolytic development.

## 3. β3 Adrenergic Receptors

The β3 adrenergic receptor gene encodes a single 408–amino acid receptor protein that belongs to the G-protein-coupled receptor (GPCR) family, with three intracellular and three extracellular loops [53]. The N-terminal region is extracellular and glycosylated; the C-terminus is intracellular. β3 adrenergic receptors differ from β1 and β2 receptors, with dissimilarities clustered in the third intra-cytoplasmic loop and C-terminal tail where the β3 adrenergic receptor lacks the consensus sequences for phosphorylation by βARK (β adrenergic receptor kinase, also known as GRK2) [54]. βARK mediates β-arrestin recruitment, followed by internalization and homologous β2 desensitization. Because of the lack of a consensus sequence for βARK, the β3 adrenergic receptor does not demonstrate refractoriness.

The β3 adrenergic receptor was cloned in 1989 [53] and shown to mediate lipolysis [55] and thermogenesis [56] in adipose tissue. β3 adrenergic receptors have been well studied in the bladder, and the β3 agonist mirabegron (Myrbetriq™) is available for treatment of overactive bladder [57]. β3 adrenergic receptors are also expressed in other smooth muscles including myometrium [58]. The treatment of human myometrial strips in tissue bath experiments revealed relaxation to β3 adrenergic receptor agonists SR59119A [58] and BRL37344 [59]. However, the transduction of β3 adrenergic receptor stimulation to signal relaxation is incompletely understood. Rouget et al. described the β3 adrenergic receptor as the predominant non-desensitizing [54] isoform in the pregnant uterine muscle, suggesting a doubling of expression near term [58]. 

Previous work imagined the therapeutic potential of β3 adrenergic receptor stimulation but did not explore signaling beyond assumptions that cAMP sub-served relaxation [60]. Furthermore, Croci et al. characterized the β3 adrenergic receptor-specific agonist SAR150640 in model cell systems using cAMP generation as the measure of response and examined effects in vivo with the finding that β3 adrenergic receptor agonist reduced myometrial contractions in conscious non-pregnant cynomolgus monkey and confirmed the failure of β2 adrenergic receptor stimulation by salbutamol to do so [60]. While these data are encouraging for β3 adrenergic receptors serving as therapeutic targets, they do not explain the mechanism(s) underlying β3 adrenergic receptor-stimulated relaxation. The data available so far underscore the failure of cAMP to explain the relaxant effects of these compounds. 

### 3.1. β3 Adrenergic Receptor Pharmacology

The β3 adrenergic receptor has high affinity and potency for selective agonists such as mirabegron, vibegron, solabegron, and ritobegron. β1 adrenergic receptor and β2 adrenergic receptor antagonists such as nebivolol, bucindolol, and pindolol are full or partial agonists of β3 adrenergic receptor, and propranolol is a non-selective low-affinity antagonist. To date, no comprehensive pharmacological studies have been performed in human myometrium, particularly in disparate states of pregnancy. Indeed, the fact of classical β-adrenergic receptor expression in the myometrium is an immediate curiosity given that there is no convincing evidence that these receptors are activated by circulating catecholamines under normal physiological conditions near term. In fact, circulating levels of epinephrine and norepinephrine are low during pregnancy [61]. Furthermore, the pregnant uterus becomes denervated during pregnancy with no evidence of adrenergic nerve fibers [62], suggesting that the regulation of the myometrium follows from local signaling alone. The notion that GPCRs can exist in multiple conformational states, some of which enjoy an active conformation in the absence of an agonist, is well accepted pharmacology [63]. The relevance of this question extends to a basic understanding of pregnancy physiology and is thus of fundamental interest. Notwithstanding the answer, β3 adrenergic receptor stimulation leads to the relaxation of the myometrial smooth muscle.

### 3.2. β3 Adrenergic Receptor Signaling

The β3 adrenergic receptor couples to both G_s_- and G_i_-receptor-transducing proteins depending on cell type (Figure 1). It cannot be overstated that despite the ability of the β3 adrenergic receptor to couple to G_s_ to stimulate cAMP generation and activation of its cognate kinase PKA, this pathway alone is unable to effectively relax the preterm laboring myometrium, as evidenced by the failure of β2 agonists as tocolytics. The only rational explanation for why one receptor coupled to cAMP accumulation fails as a tocolytic and another such as the β3 adrenergic receptor succeeds would be the compartmentation of cAMP signaling such that not all cAMP activates all PKA. While the first demonstration of cAMP compartmentation at the sub-cellular level in muscle was demonstrated by us in 1983 in cardiomyocytes [18], we find no evidence for agonist-specific cAMP action in myometrium, while there is evidence for the agonist-specific action of cGMP in human myometrium [33]. The rational conclusion from these data is that some signaling pathway other than cAMP-PKA mediates relaxation to β3 adrenergic receptor stimulation. Although β3 adrenergic receptor activation relaxes blood vessels and tissues such as of the bladder, the coupling of β3 adrenergic receptor to intracellular effectors varies. In canine pulmonary arteries, the β3 adrenergic receptor vasodilation effect is independent of the endothelium [64] while endothelium-dependent in resistance vessels from many animal species [65], through both NOS-dependent and -independent mechanisms. If not cAMP accumulation, what then mediates β3 adrenergic receptor relaxation of the myometrium?

## 4. Src Kinase

Src derives its name from the Rous virus-encoded avian sarcoma kinase v-Src, a tumorigenic variant with a truncated C-terminus [66]. Src is a non-receptor tyrosine kinase expressed ubiquitously in human tissues [67]. Human c-Src is the best-understood member of a family of nine tyrosine kinases that regulate cellular responses to extracellular stimuli [68]. The structure of Src kinases comprises a short N-terminal segment where a myristylation site links Src to the plasma membrane, followed by a unique domain, an SH3 domain, an SH2 domain, and the kinase domain occupying most of the C-terminus [69]. Src kinases are known to be activated by increases in intracellular Ca^2+^ [70] that occur rhythmically in pregnant myometrium [71]. 

The SH2 and SH3 domains have important functions. First, they constrain the activity of the enzyme via intramolecular contacts, while proteins that contain SH2 or SH3 ligands can bind to Src SH2 or SH3 domains, activate Src, and attract the complex to specific cellular locations. Src kinases regulate ion channels. There is conflicting evidence over the effects of Src-dependent phosphorylation of large-conductance BK_Ca_ channels, with inhibition being reported in rat aorta [72] and activation being reported in rat vascular muscle, which also promotes the association of the channel with integrins [73]. The reason for these discrepancies is unclear but likely relates to differences in smooth muscle sub-types and underscores the imperative that we study therapeutic questions in human tissues. Cao et al. demonstrated that conserved proline-rich motifs in the third intracellular loop and C-terminus of the β3 adrenergic receptor directly recruit Src in an agonist- and PTX-sensitive (Gi-mediated) manner. These authors showed that the interaction occurs through the SH3 domain of Src [74]. These findings support our hypothesis that uterine myocyte β3 adrenergic receptor (β3AR) coupling to Gi acquires ligand-induced tyrosine kinase activity by means of direct recruitment of Src kinase to activate BK_Ca_. Electrophysiological recordings support the notion that β3ARs activate myometrial BK_Ca_ [59]. The activation of BK_Ca_ is consistent with myometrial hyperpolarization and quiescence [22].

## 5. Connexin 43

Gap junction channels (GJCs) are composed of multi-pass transmembrane connexins [75] with four transmembrane domains (Figure 2) linked by one intracellular and two extracellular loops with N- and C-terminal cytoplasmic domains. Cx43 hexamers (hemichannels, HCs) form in the secretory pathway and are trafficked to the cell surface where they can act as stand-alone channels with robust conductance (>200 pS) [76] or dock with HCs in appositional membranes to form intact GJCs. Both states are regulated and critical to myometrial function. HC activity promotes quiescence, likely by the release of ATP [77], thought to potentially activate ATP potassium (K_ATP_) channels and may have antiinflammatory activity, while GJCs promote contraction by forming cell-to-cell communication that provides for electromechanical signal propagation of particular importance in myometrium where nervous regulation is absent. There are 21 members of the human connexin family, and Cx43 is abundant in myometrium. Interestingly, NO donors S-nitrosate Cx43 [78]. Based on the highly resolved crystal structure of Cx26 (used to extrapolate to other connexins), extracellular regions E1 and E2 have three cysteines each, and it is thought that these are sites of disulfide bond formation between connexins in appositional membranes linking HCs into GJCs. We propose that S-nitrosation of one or more cysteines in these domains prevents GJC formation, while S-nitrosation of S271 in the intracellular C-terminal tail regulates O-phosphorylation at serine 368 (pS368), preventing GJC assembly [79] and activating the HC [80]. Thus, S-nitrosation of Cx43 maintains Cx43 in the HC state. GJCs form following the loss of S-nitrosation and support coordinated electrical coupling and contraction of the myometrium [81,82]. Dysregulation of the GSNO-mediated S-nitrosation offers a persuasive explanation of the loss of gestational relaxation and labor initiation and thus is an attractive basis on which to explain PTL.

We have found Cx43 expression to be higher in all pregnancy groups with the notable exception of PTL [83]. Dramatically increased Cx43 expression is neither necessary nor sufficient by itself to initiate labor contractions in a receptive uterus. Rather, it is the regulated formation of GJCs from extant HCs that is required and is a focus of our investigations. Increased Cx43 expression resulting in increased GJC formation in PTL may be associated with infections such as chorioamnionitis in women and with lipopolysaccharide treatment that induces labor in mice [84,85]. These data fit with the work of Lye and Mesiano [86] showing the hormonal regulation of Cx43 trafficking, although their model does not take account of membrane HC-mediated relaxation.

Cx43 undergoes phospho–dephosphoregulation. Cx43 is phosphorylated at multiple C-terminal sites [87,88] that regulate the export of the protein to the plasma membrane, oligomerization, gap junction assembly, gap junction gating, and connexin degradation [88]. Phosphorylation at serine S365 appears to regulate Cx43 hemichannel localization and half-life [89], while phosphorylation at serines 325/328/330 is thought to regulate connexin assembly into GJCs [90,91], and pS368 to prevent GJC formation. Detailing the phosphorylation of all intracellular C-terminal residues of Cx43 that reveal non-junctional properties [92,93] is beyond the scope of this review. Because changes in Cx43 phosphorylation at S368 are associated with diminished GJC formation and this is associated with its S-nitrosation, relaxation to NO is likely regulated by S-nitrosation of Cx43 [94]. Few studies of Cx43 in human myometrium appear in the recent literature, and none are instructive as to the role of either S-nitrosation or O-phosphorylation, or the interaction of both in relation to Cx43 hemichannel function in PTL. 

The recent demonstration that mirabegron stimulation of the β3 adrenergic receptor signals in part by promoting NO production, and that β3 activates Src kinase to phosphorylate tyrosine at position Y265 on Cx43 in pregnant human uterine myocytes, fits with a role for β3 agonists in tocolytic development [94]. Immunofluorescent imaging indicates that mirabegron decreases the expression of Cx43 in uterine myocytes and mediates the relaxation of uterine tissue strips over a 24 h exposure period, suggesting that mirabegron has long-lasting quiescent effects on the human myometrium. The relationship between the β3 adrenergic receptor and downregulation of the contractile-associated protein Cx43 through activation of Src kinase suggests that mirabegron may be useful in combination tocolysis (Figure 2).

## 6. Conclusions

No available medications reliably interrupt established PTL to allow an afflicted pregnancy to go to term. We have accepted the notion of 48 hrs. of efficacy for acute tocolysis with little evidence [95] and abundant anecdote. We are comfortable explaining tocolytics as providing time for neonatal steroid and to transport patients to hospital. A better goal is to prevent PTL to avoid PTB. We will not advance the treatment of PTL to prevent PTB until we generate a detailed understanding of the specific pharmacology of human myometrium, not that borrowed from efficacy in unrelated maladies in other muscles. Indeed, the fact of pregnancy alone introduces a major set of variables in applying therapeutic evidence from one application to another.

Because we still cannot meaningfully delay PTB, babies that survive very early birth experience life-long deleterious health complications. Prematurity is said to cost billions of dollars in excess medical expenses that have undoubtedly increased since first estimated by the Institute of Medicine at USD 26 billion annually in 2005 dollars [83]. Our inability to fully understand the causes of idiopathic PTB and our lack of effective tools to prevent it constitute a public health crisis.

Recent discoveries from our laboratory have innovated the field. Not only have we identified that NO-mediated human uterine smooth muscle relaxation at term is cGMP-independent [5,93], and dysfunctional in PTL, we have also found that numerous contractile-associated proteins are dysregulated in preterm myometrium, a problem that can be ameliorated with targeted monotherapies [27], combination tocolysis [83], and the repurposing of existing FDA-approved drugs. We seek to advance the field by exploring β3 adrenergic receptor-mediated effects coordinated in both myocytes and microvascular endothelial cells, resulting in a better understanding of the molecular underpinnings that drive PTL, and potentially provide for the development of novel tocolytics.

There is no good model of human PTL. With rare exception, animals in nature do not experience this phenomenon, and inducing PTB with the use of progesterone antagonists or mimicking infection with lipopolysaccharide in mice is a poor cousin to experiments in human tissue. Our success in employing PTL myometrium in our studies, while challenging, is critical to designing effective approaches to tocolysis. Studies in human pregnancy myometrium offer an obvious benefit: the ability to identify pathways and develop therapeutics in the actual target tissue. The importance of studying the problem of PTB in human tissues is further significant because pregnant women are dramatically underrepresented in drug development, and pregnant women are administered drugs “*off label*” without sufficient evidence of efficacy [96] or coordinated national reporting [97]. The known disparity in birth outcomes in African American mothers is unexplained and may be the result of dysfunctional regulation of S-nitrosation targets that might respond to β3 adrenergic receptor stimulation. Taken together, by advancing our understanding of dysregulated/unique proteins and pathways in preterm myometrium to include the β3 adrenergic receptor, we are better suited to identify effective therapies to halt PTL and delay or prevent PTB. 

Thus, focusing on traditional approaches to modulate muscle such as extracellular Ca^2+^ entry alone will not yield a viable approach to tocolytic drug development. Rather, we suggest that focusing on β3 adrenergic receptor signaling, as discussed here, in combination with entities that take advantage of unique myometrial signaling can be the basis for a combination tocolytic regimen that offers significant promise. While no studies of mirabegron for tocolysis are underway at present, it seems possible to view mirabegron as a candidate for repurposing.

## Figures and Tables

**Figure 1 biomolecules-13-01005-f001:**
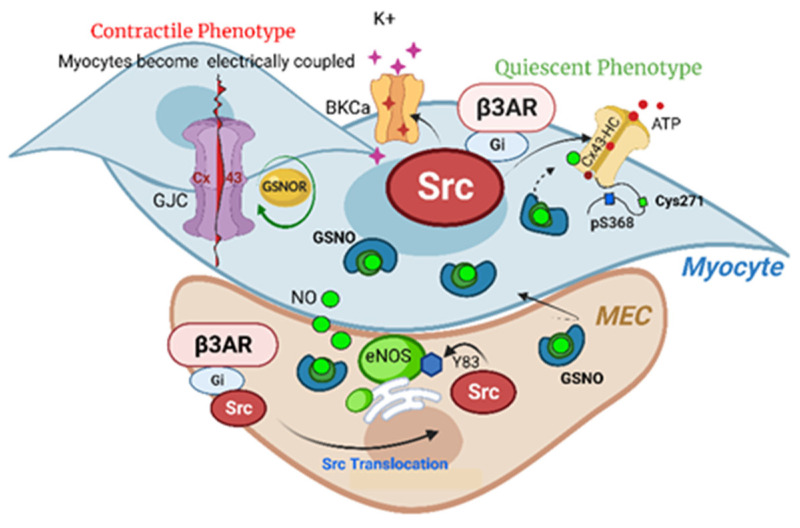
β3 Adrenergic Regulation of Myometrial Function. β3 receptors in the uterine muscle (myometrium) are present on both microvascular endothelial cells (MECs) and smooth muscle cells (myocytes). Stimulation of β3 receptors with the agonist mirabegron activates Src kinase in the MEC, leading to phosphorylation of tyrosine at position 83 (T83) of endothelial nitric oxide synthase (eNOS). eNOS-generated NO (green circles) acts on adjacent and nearby myocytes as NO or buffered as S-nitrosoglutathione (GSNO). In the myocyte, endothelial NO acts to nitrosate Cx43 at cysteine at position 271 (Cys271), promoting Cx43 phosphorylation at serine 368 (pS368) which prevents gap junction (GJC) formation and activates the hemichannel. S-nitrosation is removed by overexpression of GSNOR in preterm muscle, limiting the relaxation and promoting Cx43 GJC formation. In the myocyte, stimulation of β3 adrenergic receptors phosphorylates both the calcium-activated potassium channel BKCa and Cx43 at multiple sites. Hyperpolarization of the membrane by BKCa leads to decreased GJC activity and disassembly consistent with a quiescent phenotype. *Generated with BioRender*.

**Figure 2 biomolecules-13-01005-f002:**
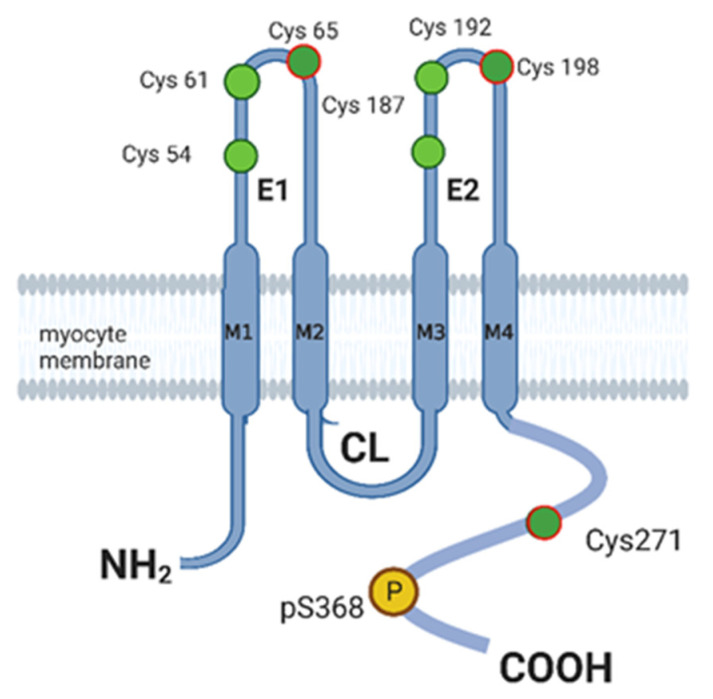
Cx43 Regulatory Sites of Putative S-Nitrosation. Each connexin (Cx) traverses the membrane with 4 spanning domains (M1-M4), 2 conserved extracellular domains (E1-E2), and 3 intracellular domains—amino and carboxyl termini and 1 variable cytoplasmic loop (CL). The regions E1 and E2 interact with the Cx43 channels of adjacent cells when forming GJCs. Cysteine (Cys) S-nitrosation (cysteine sites are in green, putative S-nitrosations are red-bordered) regulates phosphorylation (P) of serine (pS) 368 and GJC channel formation and function. *Generated with BioRender*.

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
