# Peer review of "β3 Receptor Signaling in Pregnant Human Myometrium Suggests a Role for β3 Agonists as Tocolytics"

_biomolecules, 2023, doi:10.3390/biom13061005_

Round 1
Reviewer 1 Report
Very good review, I enjoyed read it. Comprehensive, with clear point.
Some minor suggestions.
Lines 96-97 the reference is not appropriate.
Paragraph lines101-105 To my opinion it is to early to suggest β3AR signaling. It could be more effective to express it later.
Line 299-300 could be: ...Cx43 at multiple sites. Тhe dot is redundant.
References:
24. is it still in press? from 2022?
34 and 36 are the same.
48 and 49 are the same
100 and 101 are the same
Must be renumbered.
Author Response
Response to Reviewer 1
We thank the reviewer for the important suggested improvements.
Lines 96-97 the reference is not appropriate. Lorca is correctly referenced now.
Paragraph lines101-105 To my opinion it is to early to suggest β3AR signaling. It could be more effective to express it later. Moved to conclusions.
Line 299-300 could be: ...Cx43 at multiple sites. Тhe dot is redundant. Done
References:
- is it still in press? from 2022? Corrected.
34 and 36 are the same. Corrected
48 and 49 are the same. Corrected
100 and 101 are the same. Corrected
Must be renumbered. Done
Reviewer 2 Report
The author prepared a review entitled “β3 receptor signaling in pregnant human myometrium suggests a role for β3 agonists as tocolytics”, which describes the current knowledge on the mechanisms underlying human myometrium contraction/relaxation/quiescence in PTL, emphasizing the role of β3 receptor signaling in this process.
Although this is an interesting theme, and scientifically relevant to the audience of Biomolecules, there are still questions that needs to be addressed, namely:
1. Abstract should not include references
2. For the sake of clarity, the authors should carefully revise all abbreviations used in this manuscript: PTL and PTB, Cx43, NO. In some cases, abbreviations are not used, nor are refer in the first-use. Most important, β3 nomenclature is presented throughout the whole manuscript, as “β3AR” or “β3 receptor”. Please revise the nomenclature.
3. Page 5, line 197-205 – This paragraph lacks references.
4. Please reformulate sentence: Page 1, lines 43-48.
5. Page 3, line 100 - Finish sentence or give examples of “Both channel activators and inhibitors are available.”
6. Please refer to figures (1 and 2) in the text and add figure in the correct section. Example: Page 8 line 304 “Fig. 1”?
7. The authors should consider adding a Table with current treatments and its clinical drawbacks.
8. Are there ongoing clinical trials. Or current drug regimen including b3-agonists for PTL? Also, please refer to preclinical studies.
Author Response
We thank our reviewer for helping us improve our manuscript.
- Abstract should not include references. Removed.
- For the sake of clarity, the authors should carefully revise all abbreviations used in this manuscript: PTL and PTB, Cx43, NO. In some cases, abbreviations are not used, nor are refer in the first-use. Most important, β3 nomenclature is presented throughout the whole manuscript, as “β3AR” or “β3 receptor”. Please revise the nomenclature. Done
- Page 5, line 197-205 – This paragraph lacks references. Done
- Please reformulate sentence: Page 1, lines 43-48. We do not see an issue with this text in this section; we are hopeful that we did not misunderstand the review comment.
- Page 3, line 100 - Finish sentence or give examples of “Both channel activators and inhibitors are available.” Done
- Please refer to figures (1 and 2) in the text and add figure in the correct section. Example: Page 8 line 304 “Fig. 1”? Done
- The authors should consider adding a Table with current treatments and its clinical drawbacks. Thank you for the suggestion. We do not think it advisable to offer a table of treatments for preterm labor. The differences among acceptable practice are geographically disparate and nothing is effective as discussed in the manuscript. A table would appear as an indorsement of sorts that is beyond our purview. We have offered additional text to cover the reviewers concern, including consideration of point 8 below.
- Are there ongoing clinical trials. Or current drug regimen including b3-agonists for PTL? Also, please refer to preclinical studies. No clinical trials are underway and our preclinical studies and those we quote are those we are aware of.
Round 2
Reviewer 2 Report
-